# Crystal Structure of Staphopain C from *Staphylococcus aureus*

**DOI:** 10.3390/molecules28114407

**Published:** 2023-05-29

**Authors:** Malgorzata Magoch, Alastair G. McEwen, Valeria Napolitano, Benedykt Władyka, Grzegorz Dubin

**Affiliations:** 1Malopolska Centre of Biotechnology, Jagiellonian University, 30-387 Krakow, Poland; m.magoch@gmail.com (M.M.);; 2Department of Analytical Biochemistry, Faculty of Biochemistry, Biophysics and Biotechnology, Jagiellonian University, 30-387 Krakow, Poland; 3CNRS, INSERM, Université de Strasbourg, IGBMC UMR 7104–UMR-S 1258, F-67400 Illkirch, France

**Keywords:** *Staphylococcus aureus*, staphopain, ScpA2, virulence factor, cysteine protease, X-ray crystallography

## Abstract

*Staphylococcus aureus* is a common opportunistic pathogen of humans and livestock that causes a wide variety of infections. The success of *S. aureus* as a pathogen depends on the production of an array of virulence factors including cysteine proteases (staphopains)—major secreted proteases of certain strains of the bacterium. Here, we report the three-dimensional structure of staphopain C (ScpA2) of *S. aureus*, which shows the typical papain-like fold and uncovers a detailed molecular description of the active site. Because the protein is involved in the pathogenesis of a chicken disease, our work provides the foundation for inhibitor design and potential antimicrobial strategies against this pathogen.

## 1. Introduction

*Staphylococcus aureus* is a Gram-positive bacterium responsible for an array of human and livestock infections [1]. The increasing antibiotic resistance of *S. aureus* presents a major clinical challenge, driving intense research into its physiology [2]. In humans, the pathogen is associated with serious nosocomial and community-acquired infections. It colonizes the skin, nares and mucosal membranes and results in a wide spectrum of symptoms, ranging from skin diseases to lung inflammation, bacteremia/sepsis, endocarditis, pneumonia and osteomyelitis [3]. Apart from being a serious threat to public health, *S. aureus* is also a major cause of animal diseases including skeletal infections of poultry, resulting in a serious impact on food production. Along with the direct effects on animal health, animals act as a reservoir for staphylococcal transmission to humans [4,5]. A major contribution to the success of *S. aureus* as a pathogen resides in a large number of extracellular virulence factors that manipulate the host’s innate and adaptive immune responses [6]. Several lines of evidence suggest that the secreted proteases participate in staphylococcal virulence [2,7]. Staphylococci produce proteases of three catalytic classes, which include serine proteases, metalloproteases and cysteine proteases (staphopains) [8]. Human strains secrete two papain-like cysteine proteases (staphopains A and B), whereas avian strains express staphopain C (ScpA2) [9]. ScpA2, an enzyme distinct from both human homologs, was first reported in 1967 in a high-protease-producing strain, CH-91, obtained from a chicken suffering from edematous and necrotic dermatitis [10]. Further studies conducted by Takeuchi and colleagues [11] involved the purification of the enzyme and characterization of some of its basic properties. The authors also demonstrated that the gene encoding ScpA2 was characteristic of the CH-91 strain [12]. Later findings suggested that ScpA2, as a poultry-specific virulence factor, is directly involved in the pathogenesis of chicken diseases [13]. Subsequent genetic analyses associated the high level of virulence in the chicken embryo model with the gene encoding a cysteine protease [14]. In addition, biochemical studies demonstrated that staphopain C is efficiently inhibited by alpha-1-antichymotrypsin in human plasma, whereas it remains fully active in chicken blood plasma, providing an explanation for the species-specific distribution of ScpA2 only in the avian strains [9].

Staphopain C is a typical cysteine (thiol) proteinase. Cysteine proteinases are characterized by a molecular weight of approximately 21–30 kDa. They are often synthesized as inactive proenzymes, activated by limited proteolysis. Cysteine proteinases participate in numerous physiological and pathological processes. The name of the group relates to the catalytic function of the thiol moiety (–SH) of the catalytic cysteine—a central element of the active site catalytic triad. Within the triad, the cysteine is deprotonated by an adjacent basic side-chain histidine residue. In such a state, the cysteine catalyzes the hydrolysis of peptide bonds via nucleophilic attack on the substrate carbonyl carbon. A tetrahedral intermediate is stabilized at the oxyanion hole and breaks up, releasing a product containing a new amino terminus. Subsequently, the thioester bond is hydrolyzed to produce a new carboxy terminal containing a product, regenerating the enzyme [15].

In the present study, we determined the three-dimensional crystal structure of ScpA2 at 1.58 Å resolution. Our structure shows the typical papain-like fold and discloses the detailed molecular characteristics of the active site, thereby uncovering the molecular basis of the substrate recognition and characterization of the active site for future inhibitor design.

## 2. Results and Discussion

### 2.1. Expression and Crystallization of ScpA2

Staphopain C was obtained from the culture supernatant of *S. aureus* strain CH-91 [11]. Total extracellular proteins were precipitated with ammonium sulfate, and ScpA2 was obtained via subsequent ion exchange and gel filtration chromatography. The final preparations were characterized by at least 95% purity, as analyzed using sodium dodecyl sulfate polyacrylamide gel electrophoresis (SDS-PAGE; Figure 1). The protein was concentrated to 20 mg/mL and subjected to crystallization screening in a sitting drop vapor diffusion setup. The initial crystallization conditions were further optimized, allowing us to repeatably obtain high-quality, X-shaped crystals (see Appendix A) of ScpA2 suitable for the structure determination.

### 2.2. Overall Crystal Structure of ScpA2

ScpA2 crystals diffracted up to 1.58 Å. Initial attempts to solve the data via molecular replacement using residues 229–399 of the AlphaFold [16,17] model L7PGC4 as a search model yielded clear replacement results and a sharp map showing distinct features characteristic of ScpA2, but the model failed to refine the structure in terms of satisfactory R_free_ values. Data quality assessment using Xtriage [18] showed that the crystals were tetartohedrally twinned, suggesting a (pseudo)merohedral twinning with four twinned crystal domains [19]. Applying the relevant twinning laws, we found that the crystals belonged to the space group P3_1_ with unit cell parameters of a = b = 103.15 Å and c = 51.21 Å (α = β = 90°, γ = 120°). Molecular replacement was carried out using the AlphaFold model as a search model. Three molecules of ScpA2 were found in the asymmetric unit characterized by a solvent content of 54%. The obtained electron density maps were of excellent quality, and the majority of the enzyme was well-defined. The structure was refined to a final R_work_ value of 9.06% and R_free_ of 11.3%. The final model of ScpA2 consists of a total of 519 amino acid residues (173 for each monomer), 413 water molecules and 5 chlorine ions. The three molecules in the asymmetric unit are highly similar (root mean square deviation (RMSD) of 0.09 Å) (Figure 2a). Slight differences are confined to the C-terminal residues and the small loop spanning residues Leu129 to Gly132. The electron density around residue 101 is missing, and the residue is modeled as aspartic acid in reference to sequence no. Q8RJP3 deposited in the UniProt database. The relevant omit map is provided in the Appendix A. The lack of electron density of the side chain is attributed to the flexibility (multiple conformations) of the side chain.

The X-ray structure of ScpA2 shows the typical fold of the proteases belonging to the papain family. The fold consists of two domains roughly equal in size. In papain-family proteases, these domains are generally referred to as L (left) and R (right), and such nomenclature will be used in this work (Figure 2b). The domains are separated by a V-shaped cleft that contains the active site harboring the conserved catalytic triad (Cys23, His119, Asn140) residues. The L domain is composed of three α-helices (α1, α2 and α4). Helices α2 and α4 are connected through an extended loop, which includes an additional small helix (α3). The helix α1 carries the nucleophilic cysteine (Cys23) of the catalytic triad at its amino terminus. The R domain is composed of a distorted eight-stranded antiparallel β-barrel. The connection between β1 and β2 is extended and includes a short helix (α5). The R domain contains the other two residues of the catalytic triad, His119 on β2 and Asn140 in the loop connecting β4 and β5. The L and R domains interact through an extended amphipathic interface stabilized by numerous hydrogen bonds as well as by hydrophobic contacts. ScpA2 shows 80% sequence identity with staphopain A (ScpA) and 51% sequence identity with staphopain B (SspB) (Table 1). The structural alignment of SspB (PDB code: 1Y4H) and ScpA (PDB code: 1CV8) with the structure of ScpA2 determined here is characterized by RMSDs of 1.3 Å and 0.4 Å (according to Dali server [20]), respectively, over 173 residues, in agreement with the significant sequence homology of the enzymes (Figure 3). The main differences arise in the orientation of the long loop connecting the β3 and β4 strands, the short loop connecting the β4 and β5 strands and the loop containing the small helix, which bridges helices α2 and α4 and the configuration of the β1 strand at its N-terminus (see Appendix A).

### 2.3. Active Site

The well-described catalytic mechanism of papain is shared by all the homologs commonly known as papain-like cysteine peptidases. The basic features of the mechanism include the formation of a covalent intermediate, the acyl-enzyme, resulting from the nucleophilic attack of the active site thiol group on the carbonyl carbon of the scissile amide or the ester bond of the substrate.

In ScpA2, the active site is located at the interface of the L and R domains. The α1-helix contained within the N-terminal part of the protein (L domain) carries a nucleophilic cysteine residue (Cys23)—a key structural element of the catalytic site. The C-terminal R domain built around an eight-stranded antiparallel pseudobarrel contributes the catalytic His119 and Asn140. Cys23 forms a thiolate–imidazolium ion pair with His119 (3.5 Å). The histidine acts as a proton acceptor, making the cysteine a permanent, negatively charged nucleophile, while the third residue of the triad (Asn140) plays an important role in positioning His119 in a relevant orientation by forming a hydrogen bond (2.7 Å) between the amide oxygen of its side chain and the Nε2H of His119 (Figure 4). The asparagine side chain is buried in a region of the enzyme composed mainly of hydrophobic residues: Phe141, Val158, Trp139 and Trp142, which shield the Asn140–His119 hydrogen bond from the solvent. An important feature of the Asn140–His119 interaction is that the hydrogen bond is approximately colinear with the His119 Cβ–Cγ bond, allowing the rotation of the imidazole ring about the Cβ–Cγ bond without disruption of the Asn140–His119 hydrogen bond. As such, the role of Asn140 is to orient the His119 side chain in the optimum position for the consecutive steps of catalysis, and to stabilize the thiolate–imidazolium ion pair [23,24]. Recent data suggest an important role of Trp142 in the generation of the nucleophilic character of the catalytic dyad. Decreased shielding of the thiolate–imidazolium pair from the solvent by the movement relative to Trp142 would enhance the nucleophilic character [25,26].

Another important feature of the catalytic machinery of cysteine proteases is the oxyanion hole, a structure stabilizing the tetrahedral transition state of the thioacyl-enzyme catalytic intermediate formed. In ScpA2, the oxyanion hole is composed of two hydrogen bond donors: the Gln17 side chain, and the backbone NH of catalytic Cys23. It has been demonstrated that the Gln17Ala and Gln17Ser mutants of papain lead to a reduction in its activity, confirming the role of this side chain in enhancing the rate of catalysis of the enzyme [15,27].

### 2.4. S2 Subsite Determines Substrate Specificity

Although the overall topological features and mechanism of catalysis among known staphopains are conserved, different side chains lining the active site cleft may define different substrate specificities for individual members. Following the Schechter and Berger notation, the active site encompasses specific subsites (Sn and Sn’) that accommodate consecutive residues of the substrate (Pn and Pn’) [28]. The S1 subsite accommodates a substrate residue encompassing the newly formed C-terminus of the cleaved polypeptide chain. The S1’ subsite accommodates the residue containing a newly formed N-terminus. It is widely accepted that for papain-like cysteine proteases, the S2 subsite is the primary specificity determinant—the enzymatic specificity of cysteine proteases of the papain family primarily depends on P2/S2 interactions [29]. In staphopain C, the S2 subsite constitutes a hydrophobic pocket lined with Pro66, Met69, Lue95 and Ala117, all of which are conserved among the three best-characterized homologs (ScpA, SspB and ScpA2) [2]. This S2 pocket enables the processing of substrates with a hydrophobic side chain at the P2 site, consistent with the information available on the substrate preference of ScpA2 [30]. Despite the overall similar character of the S2 specificity pockets of the compared enzymes, certain significant differences are also visible, which fine-tune the specificity. It was shown by Fillipek and coauthors [31] that within the complex of SspB with a proteinaceous, substrate-mimicking inhibitor, namely staphostatin B, a hydrogen bond is formed between the main chain of Ile in position P2 of the inhibitor and Thr64 of the protease. Threonine is specifically present only in SspB (while ScpA and ScpA2 accommodate leucine in the discussed position), which may explain the differences in the substrate preferences of the respective enzymes. In addition, our structure contains a unique Asn167 residue where the two homologs contain Tyr167 in the corresponding position. We speculate that this particular residue has a role in fine-tuning the specificity of ScpA2. The previously solved structures of ScpA2 homologs were determined in the presence of substrate-mimicking inhibitors that occupy the active site of the enzyme (complex of staphopain A [32] with the epoxide inhibitor E-64) or the inactive form (Cys243Ala mutant with staphostatin B [31]). Our structure is the first to document the active form of apo-enzyme. The comparison with inhibitor-containing/inactive homologs reveals no notable differences in the arrangement of the catalytic residues (Cys23, His119 and Asn140), indicating that the catalytic machinery is pre-oriented independently of substrate binding. Nonetheless, differences are noted in the loop preceding the catalytic histidine (between β3 and β4 of the R domain). Because the orientation of the loop varies in the three compared structures, it is difficult to distinguish the substrate-binding effects from the general flexibility of the loop regions. Nonetheless, the loop may have importance in determining the substrate specificity of the enzymes.

## 3. Materials and Methods

### 3.1. Expression and Purification of ScpA2

Staphopain C was obtained from the culture supernatant of *S. aureus* strain CH-91. The cells were harvested via centrifugation (10,000× *g* for 20 min at 4 °C), and total extracellular proteins from the supernatant were precipitated with ammonium sulfate at 80% saturation (561 g/L). The precipitated proteins were collected via centrifugation. The obtained pellets were dissolved and dialyzed against 20 mM phosphate buffer, pH 8.0, and purified on Q Sepharose FF using a linear gradient of 0–0.5 M NaCl in the same buffer. Further purification was carried out via size-exclusion chromatography (Superdex 75, GE Healthcare, Chicago, IL, USA) in 5 mM Tris-HCl and 50 mM NaCl, pH 8.0. Fractions containing the protein of interest were identified using SDS-PAGE, and only fractions of at least 95% purity were used (Figure 1). Purified samples were used for crystallization. The protein concentrations were determined with a BSA assay (Sigma, St. Louis, MO, USA).

### 3.2. Crystalization of ScpA2

ScpA2 was concentrated to ~20 mg/mL and used directly for the crystallization screening that was performed using the sitting drop vapor diffusion method with commercially available crystallization kits. Briefly, 1 µL drops of protein solution were mixed with an equal volume of the reservoir solution, and then the plates were sealed and incubated at 20 °C. The initial crystals appeared after 3–5 days in 1.5 M ammonium sulfate, 0.1 M Tris-HCl pH 8.5 and 15% glycerol and were further optimized. Diffraction-quality crystals were obtained from 1.5 M ammonium sulfate, 0.1 M Tris-HCl pH 8.5 and 12% glycerol. The crystals were cryo-protected in mother liquor supplemented with 20% glycerol and flash-cooled in liquid nitrogen.

### 3.3. Data Collection and Crystal Structure Solution

Data were collected from a single crystal on the BL14.2 beamline of the BESSY II electron storage ring (Berlin-Adlershof, Germany) at the Helmholtz-Zentrum Berlin (HZB). The data were processed and scaled with autoPROC [33,34,35,36,37,38] in space group P622 with the unit cell dimensions a = b = 103.16 Å, c = 51.21 Å, α = β = 90° and γ = 120°, with an estimated 1 molecule in the asymmetric unit via Matthews’ analysis (VM = 1.99 Å^3^·Da^−1^, 38.3% solvent) [39]. In total, 5% of the reflections were selected for the test set and assigned to thin resolution shells. The clear molecular replacement solution in PHASER [40] using residues 229–399 of the AlphaFold [16,17] model L7PGC4 as a search model did not refine the structure. Analysis of the data with XTRIAGE from the PHENIX suite [41] indicated twinning. The data were reprocessed with autoPROC using lower-symmetry space groups (P6, P321, P312, P3 and C222), and molecular replacement was attempted. A good solution (TFZ = 51.7, LLG = 3511) was found in space group P3_1_ (a = b = 103.16 Å, c = 51.21 Å, α = β = 90°, γ = 120°) with 3 molecules in the asymmetric unit (VM= 2.68 Å^3^·Da^−1^, 54.1% solvent). No complete solutions were found in any of the other space groups. Since the data showed tetartohedral twinning [19], the structure was refined using 4 twin domains with the following twin fractions: H, K, L (29.3%), H, H, -L (20.8%), -K, -H, -L (20.6%) and -H, -K, L (29.3%). The early stages of the refinement were performed in PHENIX.REFINE [42], with iterative manual rebuilding performed in COOT [43]. The final stages of the refinement were performed in REFMAC [44] in the CCP4 suite [37]. The model quality was assessed using MolProbity [45]. The data collection and refinement statistics are summarized in Table 2.

## 4. Conclusions

Our research provides the high-resolution crystal structure of staphopain C (ScpA2)—a poultry-specific virulence factor from the strain of *Staphylococcus aureus* that is known to cause skeletal infections in chickens, thus having a significant impact on agriculture. Our structure shows the typical papain-like fold and uncovers a detailed molecular description of the substrate recognition pocket and the catalytic machinery of the protease. Elucidation of the structure of the protease offers the opportunity to target its active site with low-molecular-weight inhibitors, providing a possible strategy against staphylococcal infections in chickens.

## Figures and Tables

**Figure 1 molecules-28-04407-f001:**
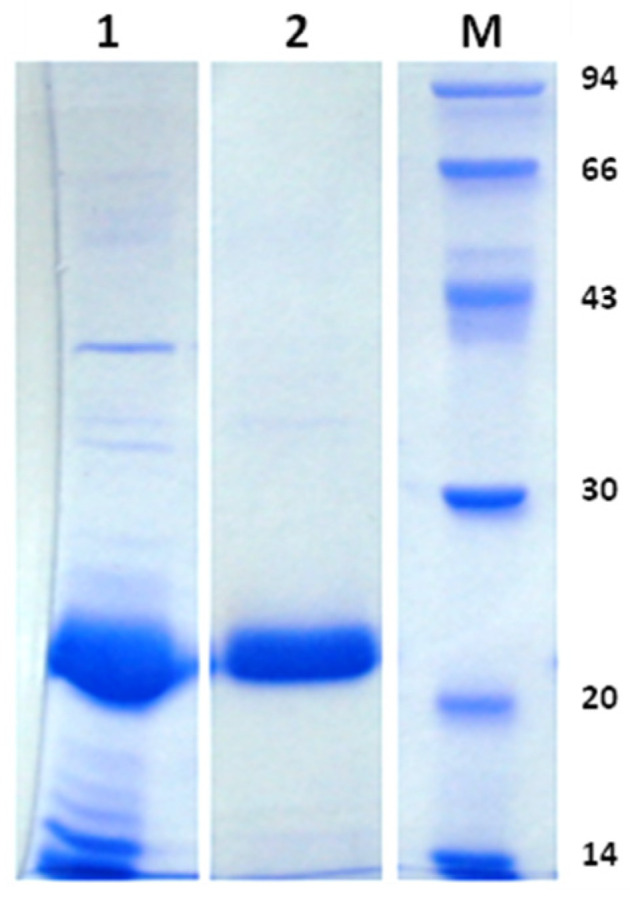
Sodium dodecyl sulfate polyacrylamide gel electrophoretic (SDS-PAGE) analysis of staphopain C (ScpA2) preparations used in this study. Lane 1—proteins contained in the total culture supernatant after 15 h of culture. Line 2—staphopain C (ScpA2) after the last purification stage. This preparation was used for crystallization. Lane M—molecular weight marker (14 to 94 kDa, as indicated).

**Figure 2 molecules-28-04407-f002:**
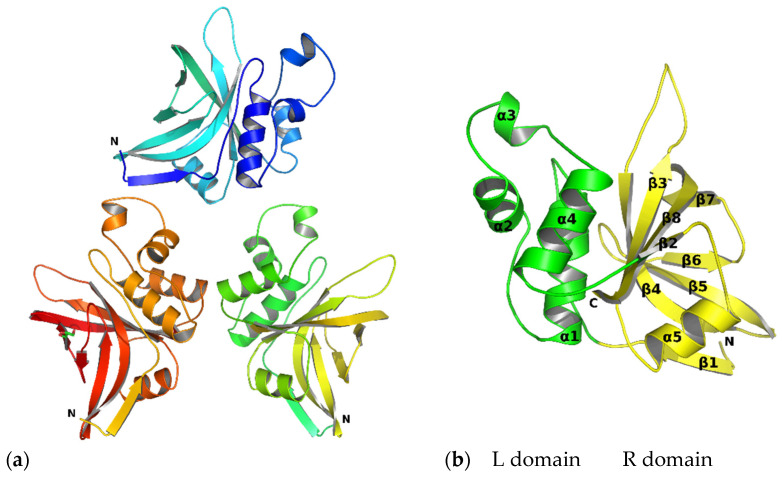
The crystal structure of staphopain C. (**a**) Contents of the asymmetric unit (PDB code: 8OIG). (**b**) Representative monomer with R (yellow) and L (green) domains. The secondary elements are numbered. All images are ribbon diagrams.

**Figure 3 molecules-28-04407-f003:**
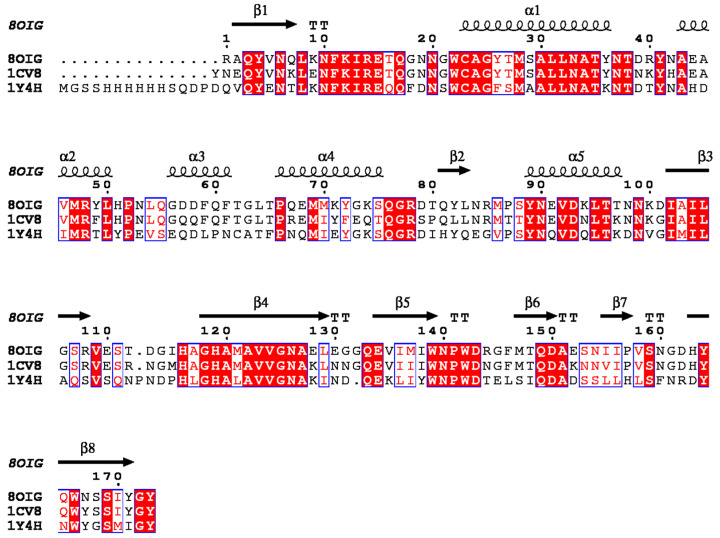
Sequence similarity among cysteine proteases secreted by *Staphylococcus aureus*. The figure shows the amino acid sequence alignment of staphopains with known crystal structures. PDB codes are indicated. 8OIG—*Staphylococcus aureus CH-91* (structure characterized in this study); 1CV8—*Staphylococcus aureus*: staphopain A; 1Y4H—*Staphylococcus aureus*: staphopain B. The conserved residues are highlighted with a red background, while homologous residues are depicted in red fonts and blue boxes. The secondary structure elements are indicated above the alignment. The figure was prepared with ENDscript [21], which uses DSSP to identify the secondary structure elements [22].

**Figure 4 molecules-28-04407-f004:**
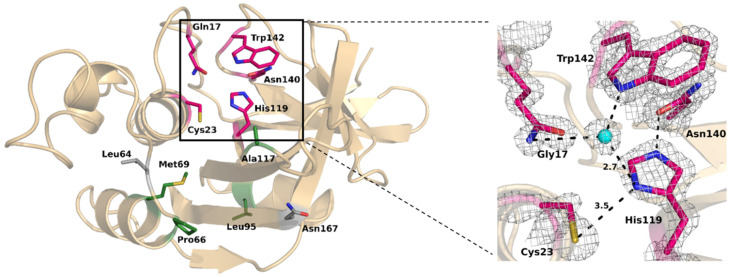
The catalytic site of ScpA2 as revealed by the crystal structure. Catalytic residues are highlighted in the thick-stick model (pink). The residues lining the P2 specificity subsite are highlighted in the thin-stick model (green), and the residues highlighted in gray (Leu64 and Asn167) might play a further role in enzyme specificity. The hydrogen bonds are depicted as dotted lines. The water molecule is shown as a blue sphere. The 2|Fo|-|Fc| electron density map at 1.58 Å resolution is contoured at 1.2 σ around the catalytic residues.

**Table 1 molecules-28-04407-t001:** Closest structural homologs. Results from a Dali search with the staphopain C (PDB 8OIG) structure.

PDB ID	Z-Score	RMSD [Å]	Sequence Identity [%]
1CV8 (staphopain A)	34.0	0.4	80
1Y4H (staphopain B)	31.1	1.3	51

**Table 2 molecules-28-04407-t002:** Crystallographic data collection and refinement statistics. The values in parentheses refer to the highest-resolution shell.

Space Group	P3_1_
Cell Parameters	
a, b, c (Å)	103.14, 103.14, 51.20
α, β, γ (°)	90, 90, 120
Wavelength (Å)	1.05
Wilson B factor (Å^2^)	11.07
Resolution range (Å)	51.57–1.58 (1.64–1.58)
Completeness (%)	97.51 (99.95)
R_merge_ (%)	4.69 (15.03)
R_meas_ (%)	5.43 (17.60)
R_pim_ (%)	2.72 (9.06))
CC1/2 (%)	99.7 (91.7)
Observed reflections	307,174 (30,350)
Unique reflections	80,816 (8329)
I/sigma (I)	25.12 (9.86)
Average multiplicity	3.8 (3.7)
**Refinement**	
Resolution (Å)	1.58
No. of reflections used	81,079 (8333)
R_factor_ (%)	9.01 (20.21)
R_free_ (%)	11.32 (20.38)
Average B factor (Å^2^)	11.12
Protein	9.85
Ligands	27.69
Water	25
RMSD from Ideal Values	
Bond length (Å)	0.013
Bond angles (°)	1.85
Ramachandran Statistics (%)	
Most favored regions	98.44
Additionally allowed regions	1.56
Content of the Asymmetric Unit	
No. of protein molecules/residues/non-H atoms	3/519/4159
Ligands	17
No. of solvent molecules	413

## Data Availability

No new data were created or analyzed in this study. Data sharing is not applicable to this article.

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
