# Peer review of "Crystal Structure of Staphopain C from Staphylococcus aureus"

_molecules, 2023, doi:10.3390/molecules28114407_

Round 1

Reviewer 1 Report

This paper reported the three-dimensional structure of the staphopain C (ScpA2) of Staphylococcus aureus and a detailed molecular description of the active site. However, the work lacks sufficient innovation and logic. Therefore, we recommend that the paper should not be accepted.

Some of the problems in this paper are listed as follows:

1. The literal descriptions exists serious problems such as inaccurate descriptions and incomplete sentences (e.g. the description in line 156).

2. The diagrams and charts do not correspond in this paper, and the corresponding diagrams are not indicated in the text description.

3. Part of the description in the results and discussion of this paper lacks the support of experimental results, such as Part 2.1 and “the electron density maps” in line 84.

4. The writing of this paper is extremely irregular, with inconsistencies in font format (e.g. 154-156 lines) and the format of references (e.g. lack of page numbers and magazine names).

Author Response

Reviewer 1

In the general section of the review, Reviewer 1 states that “… the work lacks sufficient innovation and logic.”, however without providing any detailed comments. The innovation of our work is in providing a structure of an enzyme which structure was not known prior to our study. The logic of our work is relatively simple – we purify a virulence factor of an avian pathogen and characterize its structure with the long-term view of facilitating inhibitor design – development of small molecules of potential therapeutic value. The workflow and conclusions are clearly delineated in the manuscript and in the absence of detailed reviewer comments it is difficult to address the above general reservation.

Regarding the detailed issues listed by Reviewer 1, all were addressed in the revised version as follows:

  • The literal descriptions exists serious problems such as inaccurate descriptions and incomplete sentences (e.g. the description in line 156).

We have thoroughly revised the entire manuscript to avoid inaccurate descriptions and incomplete sentences. Among others, the description in line 156 was corrected.

  • The diagrams and charts do not correspond in this paper, and the corresponding diagrams are not indicated in the text description.”

The references to diagrams and charts in the original version of the manuscript were indeed incomplete and incorrect. The references are now corrected in the revised version according to the reviewer suggestion.

  • Part of the description in the results and discussion of this paper lacks the support of experimental results, such as Part 2.1 and “the electron density maps” in line 84.

Part 2.1 refers to the protein purification. We stated the purification method and referred to the original work of Takeuchi and colleagues (Vet. Microbiol. 67, 195-202) who devised the method. Indeed, no experimental data was provided which is now corrected in the revised version (Figure 1 in the revised manuscript).

Line 84 – We made an interpretative mistake here and we thank the reviewer for pointing this out. We have mistakenly interpreted a truncated electron density of aspartic acid at position 101 as glycine, but the sequence clearly indicates aspartic acid at this position. The lack of electron density of the sidechain should be attributed to flexibility (multiple conformations) of the sidechains and not to mutation. Relevant correction was made to the text and to the pdb file (line 117 in the revised manuscript). Omit map is provided in the supporting information section of the revised manuscript, documenting that due to flexibility of the D101 sidechain, the sidechain is not covered by the density.

  • The writing of this paper is extremely irregular, with inconsistencies in font format (e.g. 154-156 lines) and the format of references (e.g. lack of page numbers and magazine names).

The font format was corrected in the whole text, the format of references was modified according to the Reviewer suggestions. We have also revised the entire manuscript and removed several other inconsistencies and irregularities not directly pointed out by the reviewer.

Along the above corrections made in response to the detailed suggestions of Reviewer 1 we have thoroughly revised the entire manuscript to improve the overall clarity and adequacy of the description.

Reviewer 2 Report

In the present study, the authors have solved the structure of staphopain C, a secretory cysteine protease, and virulence factor from Staphylococcus aureus. The article is well-written, and the data are clearly presented. However, I have a few concerns as written below.

1-    Line 18- “a foundation for development of tool inhibitors” need to be rephrased.

2-    Line 64-67- “optimized according to art” .... “yield X shaped crystals” need to be rephrased. It is unclear what the authors want to say. What is art? And does the crystal shape like X?

3-    Line 85-87 – there is going to be a significant difference between electron density for aspartate and glycine. Is the density weak around D101? The authors should show the omit map around this residue.

4-    Line 105-107 – The authors should consider including a figure showing the overlay of different structures for which the comparison is being made.

5-    Line 116 – The line seems to be incomplete. Is the content of lines 116-122 part of figure legend? It is unclear and confusing.   

6-    Line 139 – reference for Figure 2 is wrong. There are two figure 2.

7-    Line 153 – catalysis should be catalytic.

8-     In the last figure showing the structure, the authors should show the omit map around the catalytically important residues.

9-    Line 220 – “optimized according to art” needs to be clarified. What is art here?

10- In conclusion, instead of two bullet points, the authors should elaborate.

Author Response

Reviewer 2

Reviewer 2 stated that “The article is well-written, and the data are clearly presented.” Additionally, the reviewer had some detailed reservations, all of which were addressed as follows:

  • Line 18- “a foundation for development of tool inhibitors” need to be rephrased.

The sentence was rephrased according to the reviewer suggestion.

  • Line 64-67- “optimized according to art” .... “yield X shaped crystals” need to be rephrased. It is unclear what the authors want to say. What is art? And does the crystal shape like X?

The initial sentence was rephrased according to the reviewer comments. Regarding the crystal shape description – the crystals were indeed shaped like letter X. To make our description clear, we have included a photo of the X shaped crystals in the revised version of the manuscript (see supporting information).

  • Line 85-87 – there is going to be a significant difference between electron density for aspartate and glycine. Is the density weak around D101? The authors should show the omit map around this residue.

We have mistakenly interpreted weak density around D101 as amino-acid substitution. The sequence clearly shows aspartic acid at this position, and it is interpreted as such in the revised version. The manuscript text and the pdb files were corrected accordingly (line 117 in the revised manuscript). Omit map is provided in the supporting information section of the revised manuscript, but due to flexibility of the D101 sidechain, the sidechain is not covered by the density at the omit map.

  • Line 105-107 – The authors should consider including a figure showing the overlay of different structures for which the comparison is being made.”

The figure showing the overlay of staphopain structures has been included in the revised version of supporting information to visualize the differences in the structures of those enzymes discussed in the manuscript.

  • Line 116 – The line seems to be incomplete. Is the content of lines 116-122 part of figure legend? It is unclear and confusing.”  

There was a formatting problem in this part of the manuscript. It has been corrected in the revised version and now the manuscript text is clearly separated from the figure legend.

  • Line 139 – reference for Figure 2 is wrong. There are two figure 2

Figures were indeed inconsistently labeled in the original version of the manuscript. The figures and tables are now renumbered, and the references are corrected in the text to point to relevant figures / tables.

  • Line 153 – “catalysis should be catalytic.

The sentence was corrected according to the reviewer comment.

  • In the last figure showing the structure, the authors should show the omit map around the catalytically important residues.

The figure was modified according to the reviewer comment. The omit map was included around the catalytically important residues (Figure 4 in the revised manuscript).

  • Line 220 – “optimized according to art” needs to be clarified. What is art here?

The referenced sentence was rewritten to improve clarity.

  • In conclusion, instead of two bullet points, the authors should elaborate.

The “conclusions” section was rewritten. The bullet points were removed, and the conclusions are now presented as a paragraph of text.

Reviewer 3 Report

Magoch et al. purified and crystallized a cysteine protease called staphopain C from a strain of S. aureus that is known to infect chickens. Cysteine proteases are known virulence factors of S. aureus and the structure of staphopain would be potentially useful in facilitating drug development or inhibitor design against this target. Overall, the structure of staphopain C is highly similar to other staphopains (staphopains A and B) and notably the structure solved in this context is in the apo form, and the authors showed that the catalytic residues were pre-oriented independently of substrate binding. Overall, this is a well-written manuscript and the discussion by the authors is well-referenced and detailed. I recommend the publication of this manuscript in its present form.

Author Response

Reviewer 3

Reviewer 3 has concluded that “(…) Overall, this is a well-written manuscript and the discussion by the authors is well-referenced and detailed. I recommend the publication of this manuscript in its present form.” The reviewer has made no detailed comments. We have nonetheless thoroughly revised the entire text of the manuscript to avoid any minor inconsistencies and omissions. 

Round 2

Reviewer 1 Report

The manuscript is well written after revision. However, several questions should be revised before accepting this manuscript for publication.

1. Part 2.3 exists twice in the manuscript, please correct.

2. The format of the references is still inconsistent. For example, the number of author names in the reference is not uniform, either all of them should be listed or an et al as in reference 2.

Author Response

Responses to the Reviewer comments:

 Reviewer 1:

After the first cycle of revision the Reviewer 1 stated that “The manuscript is well written after revision. However, several questions should be revised before accepting this manuscript for publication”

Regarding these issues listed by Reviewer 1, all were addressed in the revised version as follows:

  1. “Part 2.3 exists twice in the manuscript, please correct”.

Numbering of this session was corrected in the revised version of the manuscript.

  1. “The format of the references is still inconsistent. For example, the number of author names in the reference is not uniform, either all of them should be listed or an et al as in reference 2”.

The format of the references in the original version of the manuscript was indeed inconsistent.
The references are now corrected in the revised version according to the reviewer suggestions.